# Quality of Life in Patients with Surgically Removed Skin Tumors

**DOI:** 10.3390/medicina56020066

**Published:** 2020-02-09

**Authors:** Laura Răducu, Adelaida Avino, Raluca Purnichescu Purtan, Andra-Elena Balcangiu-Stroescu, Daniela Gabriela Bălan, Delia Timofte, Dorin Ionescu, Cristian-Radu Jecan

**Affiliations:** 1Department of Plastic and Reconstructive Surgery, Clinical Emergency Hospital “Prof. Dr. Agrippa Ionescu”, 011356 Bucharest, Romania; raducu.laura@yahoo.com (L.R.); jecan.radu@gmail.com (C.-R.J.); 2Department of Plastic and Reconstructive Surgery, Faculty of Medicine “Carol Davila” University of Medicine and Pharmacy, 020021 Bucharest, Romania; 3Department of Mathematical Methods and Models, Faculty of Applied Sciences, University Politehnica of Bucharest, 060042 Bucharest, Romania; raluca.purtan@gmail.com; 4Department of Dialysis, Emergency University Hospital Bucharest, 050098 Bucharest, Romania; stroescu_andra@yahoo.ro (A.-E.B.-S.); delia.timofte@gmail.com (D.T.); 5Discipline of Physiology, Faculty of Dental Medicine, “Carol Davila” University of Medicine and Pharmacy, Bucharest, 020021 Bucharest, Romania; gdaniela.balan@yahoo.com; 6Discipline of Internal Medicine I and Nephrology, Faculty of Medicine, “Carol Davila” University of Medicine and Pharmacy, 020021 Bucharest, Romania; doriny27@gmail.com; 7Department of Nephrology, Emergency University Hospital Bucharest, 050098 Bucharest, Romania

**Keywords:** skin cancer, squamous cell carcinoma, basal cell carcinoma, malignant melanoma, surgery, quality of life

## Abstract

*Background and Objectives*: Skin cancer is one of the most frequently diagnosed malignancies. The main goal of the therapeutic management is total excision with the prevention of recurrence and metastasis. The quality of life of the patients with skin cancer is affected by the morbidity risk, surgery, and cosmetic or functional aspects. The aim of this study was to evaluate the quality of life of patients with skin cancer prior to and post surgical intervention. *Material and methods*: We performed a prospective study on 247 patients with skin tumors. Quality of life was evaluated through an initial questionnaire that was given to all consenting patients. This was used to determine patients’ mobility, selfcare, normal activities, pain, and despair, using a five-point Likert scale. The general autoperceived health state was also recorded using a 100-point scale. The study included the responses of all patients at hospital admission, after one month of surgery, and after one year of surgery. *Results*: In patients with squamous cell carcinoma (SCC), the general health state indicator statistically significantly decreased one month after surgery and increased at one-year follow-up. In malignant melanoma (MM) patients, mobility, selfcare, normal activities, and discomfort presented a decrease in values one year after surgery, compared to the values registered at hospital admission. In patients with basal cell carcinoma (BCC), all indicators of quality of life presented an impaired value one year after surgery, after a decreasing trend. The general health state indicator statistically significantly increased one month after surgery and after one year. *Conclusions*: Surgery is one of the main steps in treating skin cancer. It has a great impact on patients’ quality of life because of pain andthe effect on mobility and normal activities. Skin cancers influence the quality of life of patients both psychologicallyand physically.

## 1. Introduction

Skin cancer is considered to be an important public health issues worldwide. The most frequently diagnosed cancers are nonmelanoma skin cancer (NMSC) and cutaneous malignant melanoma (MM) [1]. NMSC represents a group of cutaneous lesions that do not derivefrom melanocytes. The most common forms of NMSC are squamous cell carcinoma (SCC) and basal cell carcinoma (BCC) [2]. In addition, it has been described the actinic keratosis (AK), which is a cutaneous lesion that can transform into SCC. It appears especially because of excessive ultraviolet exposure [3]. MM causes up to 80% of deaths related to cutaneous malignancy, representing 16% of the diagnosed neoplasia worldwide [4]. 

In recent decades, the cornerstone of the assessment of cancer treatment efficacy was the survival rate, as well as the recurrence or complication rates. Nowadays, the focus is also on the impact of the disease on patients’ quality of life (QoL) [5]. The consequences of the malignancy on patients’ psychosocial behavior can be assessed during three phases: at diagnosis, during treatment, and during long-term follow-up. Many factors, such as emotional, physical, aesthetic, or functional concerns regarding treatment, can significantly affect morbidity, mortality, survival rates, or prognoses. Both NMSC and MM are related to high levels of distress or behavioral alteration [6], from the impact of the scars to the diagnosis of malignancy. All these can affect patients’ ability to leada fulfilling life. Moreover, up to 25% of newly diagnosed individuals with neoplasia present symptoms of depression [7].

The aim of this study was to evaluate the impact on the QoL of the patients with skin cancer prior to and post surgical intervention using the health-outcome questionnaireEQ-5D-5L (EuroQol 5 dimensions 5 levels).

## 2. Materials and Methods

We conducted a prospective study on 247 patients admitted in the Plastic Surgery Department of the Clinical Emergency Hospital “Prof. Dr. Agrippa Ionescu,” Bucharest, Romania, with skin cancer, over a period of 24 months (January 2017 to December 2018). The inclusion criteria were age >18 years, and a current diagnosis of BCC, SCC, MM, or AK, which was considered a precancerous lesion. The preoperative data comprised demographic information, smoking history, and comorbidities. The histopathological details of the tumor and adjuvant therapies were registered. All patients filled out the EQ-5D-5L questionnaire to describe their perceived QoL.

Statistical analyses were performed using the SPSS software version 19.0 (SPSS Inc., Chicago, IL, USA). For all continuous data, a preliminary statistical analysis was performed in order to test the normality (Smirnov-Kolmogorov test). For nonnormally-distributed variables and noncontinuous data, the comparisons between groups were performed using nonparametric tests (Fisher’s Exact Test, Chi-square Test, and Mann-Whitney U test), while for the normally-distributed data, Student’s *t*-test and pairedsamples *t*-test were used. In addition, correlations between different measurements were determined using the Spearman correlation coefficient. Multilinear regression models were constructed with log-transformed variables. All differences and associations were considered statistically significant if the two-sided *p*-value was less than 0.05 (95% confidence intervals (CI)).

Local ethical agreement and informed consent of the patients were obtained. The number of the document from the Ethical Commission of Clinical Emergency Hospital “Prof. Dr. Agrippa Ionescu” is 1736615, 05.08.2014.

## 3. Results

The age of the 247 enrolled patients ranged between 26 and 96 years, with amean of 68 ± 13 years. Among the enrolled patients, 142 (58%) were men and 105 (42%) were women. Moreover, 161 (65%) were from urban areas, and 86 (35%) from rural areas. For the statistical analyses, we considered the following subgroups based on the types of skin cancer: SCC group, MM group, BCC group, and AK group (Table 1).

The mean age of the patients in the MM group was statistically significantly lower than that in all the other groups (*t*-test, *p* < 0.05 for all comparisons). There was no statistically-significant difference between the subgroups regarding gender structure (Fisher’s Exact Test, *p* > 0.05). Regarding the residence area, the structures of the MM group and the AK group were significantly different from the other two groups: the patients in those groups were mostly from urban areas (Fisher’s Exact Test, *p* = 0.008 and *p* = 0.003 for MM compared to SCC and BCC, *p* = 0.014 and *p* = 0.034 for AK compared to SCC and BCC).

We also compared the localization of the tumor, the maximum tumor diameter, solar exposure, and the skin type between the four groups. Because the maximum tumor diameter does not follow a normal distribution, median and range are reported (Table 2). The maximum tumor diameter was significantly lower in BCC and AK groups, compared to SCC and MM groups (Mann-Whitney Test, *p* < 0.001), and there was no significant difference between BCC and AKgroups (Mann-Whitney Test, *p* = 0.377). The distribution of the sun exposure in SCC, BCC, and AK groups was similar, with prevalence in the “frequent” category. The amount of sun exposure in the MM group was particularly different, with equal prevalence in both categories. Skin type distribution among the patients in the four groups presented no significant differences (Chi-square Test, all *p*-values > 0.05); skin types 2 and 3 were most prevalent in all groups.

In our study, the QoL was assessed using a five-point Likert scale (from 1 to 5, 1 indicating no impairment, and 5 indicating the most severe state) for mobility, selfcare, normal activities, pain (discomfort), and despair (Table 3). The general autoperceived health state was also recorded using a 100-point scale (100 representing the best health state). The study includes the responses of all patients at hospital admission, after one month of surgery, and one year after surgery. In order to determine the parameters that affect the indicators of the QoL, we considered the correlations between several variables, such as age, gender, skin type, diabetes, chronic heart failure, tumor localization, tumor diameter, type of surgery, and the presence of other tumors. Only the statistically-significant correlations are presented.

We analyzed the QoL between the groups as follows.

Mobility: There were no significant differences between the groups.

Selfcare: After one month of surgery, there wasa statistically-significant difference between SCC and MM groups and BCC and AK groups (*t*-test, *p* < 0.001). The values in BCC and AK groups were significantly lower than those in SCC and MM groups. After one year of surgery, the value in the MM group wassignificantly lowerthanthose in the other groups (*t*-test, *p* = 0.003, *p* = 0.021, and *p* = 0.034). 

Normal activities: At hospital admission, the value in the MM group was significantly lower than those in theother groups (*t*-test, *p* < 0.001). After one month of surgery, the SCC and MM groups had significantly higher values than those in BCC and AK groups (*t*-test, *p* = 0.013). After one year of surgery, the value in the MM group was still significantly higher than those in the other groups (*p* < 0.001).

Pain/discomfort: After one month of surgery, there was a statistically-significant difference between SCC and MM groups and BCC and AK groups (*t*-test, *p* = 0.012). After one year of surgery, the value in the MM group was still significantly higher than those in the other groups (*p* < 0.001).

In the SCC group, all indicators of the QoL presented impaired values one year after surgery, after a decreasing trend (from the values registered at hospital admission), with statistically-significant differences (paired samples *t*-tests, all *p*-values < 0.05). The despair indicator slightly increased after one month of surgery, due to the impact of the histopathological result as well as the scars. The general health state indicator statistically significantly decreased after one month of surgery, and statistically significantly increased after one year (paired samples *t*-tests, *p* = 0.021 and *p* < 0.001, respectively). 

In the MM group, the mobility, selfcare, normal activities, and pain/discomfort presented a decrease in values one year after surgery, compared to the values registered at hospital admission, with statistically-significant differences in selfcare (paired samples *t*-tests, *p* = 0.007). The despair indicator statistically significantly increased after one month of surgery (paired samples *t*-test, *p* = 0.005), and remained significantly higher than at hospital admission after one year (paired samples *t*-test, *p* = 0.006), due to the impact of being diagnosed with MM. Also, the general health state significantly decreased after one month of surgery (paired samples *t*-test, *p* < 0.001), and remained lower after one year, compared to the hospital admission value (paired samples *t*-test, *p* < 0.001).

In the BCC group, all indicators of the QoL presented impaired values one year after surgery, after a decreasing trend (from the values registered at hospital admission), with statistically-significant differences, except for mobility (paired samples *t*-tests, all *p*-values < 0.05). The general health state indicator statistically significantly increased one month after surgery and after one year (paired samples *t*-tests, *p* < 0.001). 

In the AK group, all indicators of the QoL presented impaired values (1 or very close to 1) one year after surgery, after a decreasing trend (from the values registered at hospital admission), with statistically-significant differences, except for mobility (paired samples *t*-tests, all *p*-values < 0.05). The general health state indicator statistically significantly increased after one month of surgery and after one year (paired samples *t*-tests, *p* < 0.001).

## 4. Discussion

The incidence of MM and NMSC is increasing worldwide, becoming an issue for the healthcare systems due to treatment costs and morbidity [8]. The increased rate is related to the fact that in recent years, awareness of this disease has increased among patients and physicians. The number of surgical excisions of the skin tumors with confirmed histopathology has grown. Protocols have been created for the management of skin cancer for proper treatment and to prevent the disease [9]. Moreover, new studies are assessing the QoL of the patients diagnosed with skin cancer [10].

The factors that influence the QOL in patients with skin cancer are the diagnosis of the disease, surgical intervention, and scars. It must be highlighted that most of the NMSC appear on sun-exposed areas, such as the face, neck, and upper limbs [11]. These locations can lead to difficulties in the oncologic surgical excision with the best functional and aesthetic outcome [8]. In addition, the lesions can lead topain and pruritus, causing functional limitations [10]. In the case of MM, the main step of the treatment is the surgical excision, followed by the sentinel lymph node, depending on the staging of the disease [12]. Thus, the majority of skin cancers are surgically excised. In the case of positive surgical margins or recurrence, the patients must undergo a new surgical procedure. These facts change the daily routine and also have a financial impact. Postoperatively, more and more patients are complaining about scarring [10,13].

In our study, 38 patients presented SCC. The lesions appeared especially in elderly patients. In older individuals, comorbidities were associated, such as arterial hypertension and chronic heart failure [14,15]. The surgical excision was done with 0.6 cm safety margins, and the defects were covered with skin grafts or local flaps. A total of 10 patients had the lesion on the lower lip. The defects were covered with advanced flaps for a good functional result. Even if SCC of the lower lip is an invasion lesion, it does not have the same rate of metastasis as the Merkel cell carcinoma of the lower lip [16]. The QoL was affected one monthafter surgery, mostly because of the histopathological results as well asthe aesthetic and functional outcome. 

In the 168 patients who were diagnosed with BCC, the excision was performed with 0.4 cm oncologic margins. Depending on the dimension of the tumor, a primary suture was preferred, followed by local flaps and skin grafting. In patients who presented wound dehiscence, a gel with polyhexanidine was used, due to its antiseptic properties [17]. Our patients reported an excellent QoL one year aftersurgery, without local recurrence.

A study conducted by Rhee et al. (2014) on 121 patients demonstrated that the QoL of patients with SCC is more affected in comparison to those with BCC, due to the fact that SCC is a more aggressive lesion that can lead to metastasis, and has a more invasive treatment [18]. 

A total of 21 patients had AK, with good reports of the QoL after one year. Philipp-Dormston et al. (2018) demonstrated that the progression from AK to SCC is associated with a significant reduction in QoL [2]. In our study, none of the patients with AK presented SCC after one year. 

In the MM group, the QoL showed a significant decrease, with no additional increase, even after one year. The general health state had a very strong decrease after one month, and remained lower after one year, compared to the hospital admission value. The excision was performed with 0.5 cm at diagnosis, and reexcision was made with 1- or 2-cm oncologic margins, depending on the Breslow depth. The defects were covered with local flaps. Chernyshov et al. (2019) in their QoL report highlighted thatpatients with MM showed an increased physical functioning and bodily pain [19]. In addition, Newton-Bishop et al. (2014) demonstrated that the excision of the MM with more than 2 cm had a significant impact on the QoL, mostly in young patients and women [20]. The impact is higher if the melanoma is on the face, due to the scars resulting from the surgical excisionand the lymph node dissection. 

## 5. Conclusions

Skin cancer is one of the most common types of malignancies, with a rapidly increasing incidence every year, affecting patients’ QoL. Questionnaires evaluating the QoL are important and should be used by physicians as an outcome measure.

## Figures and Tables

**Table 1 medicina-56-00066-t001:** Comparative demographic data of the subgroups.

	Squamous Cell Carcinoma (SCC) Group (n = 38)	Malignant Melanoma (MM) Group (n = 20)	Basal Cell Carcinoma (BCC) Group (n = 168)	Actinic Keratosis (AK) Group (n = 21)
**Age** (mean ± SD)	74.47 ± 13.24	56.50 ± 18.53	68.67 ± 11.02	67.90 ± 16.20
**Gender**				
Male	26 (68%)	12 (60%)	91 (54%)	13 (62%)
Female	12 (32%)	8 (40%)	77 (46%)	8 (38%)
**Residence**				
Urban area	21 (55%)	18 (90%)	105 (62%)	17 (81%)
Rural area	17 (45%)	2 (10%)	63 (38%)	4 (19%)

**Table 2 medicina-56-00066-t002:** Maximum tumor diameter, solar exposure, and skin type between the subgroups.

	SCC Group (n = 38)	MM Group (n = 20)	BCC Group (n = 168)	AK Group (n = 21)
**Maximum tumor diameter** (median and range)	2.25 (0.7–4.5) cm	2.25 (0.6–9.5) cm	1.45 (0.3–6) cm	1.8 (0.3–3.8) cm
**Sun exposure**				
frequent	33 (87%)	10 (50%)	109 (65%)	17 (81%)
less frequent	5 (13%)	10 (50%)	59 (35%)	4 (19%)
**Skin type**				
1	2 (5%)	0 (0%)	3 (2%)	1 (5%)
2	15 (40%)	13 (65%)	98 (58%)	13 (62%)
3	17 (45%)	6 (30%)	66 (39%)	6 (28%)
4	4 (10%)	1 (5%)	1 (1%)	1 (5%)

**Table 3 medicina-56-00066-t003:** Comparison of quality of life between groups.

	SCC Group (n = 38)	MM Group (n = 20)	BCC Group (n = 168)	AK Group (n = 21)
**Mobility**				
Hospital admission	1.26 ± 0.6	1.60 ± 0.38	1.14 ± 0.35	1.10 ± 0.34
After one month	1.16 ± 0.37	1.75 ± 0.58	1.10 ± 0.3	1.08 ± 0.27
After one year	1 ± 0.00	1.35 ± 0.58	1 ± 0.00	1 ± 0.00
**Selfcare**				
Hospital admission	2.45 ± 0.5	2 ± 0.91	2.10 ± 0.62	2.10 ± 0.39
After one month	1.97 ± 0.54	2.05 ± 0.6	1.43 ± 0.5	1.39 ± 0.5
After one year	1.05 ± 0.22	1.40 ± 0.59	1.05 ± 0.21	1 ± 0.00
**Normal activities**				
Hospital admission	2.68 ± 0.62	2.05 ± 0.99	2.38 ± 0.66	2.29 ± 0.46
After one month	2.11 ± 0.55	2.20 ± 0.61	1.67 ± 0.73	1.46 ± 0.52
After one year	1.05 ± 0.22	1.70 ± 0.65	1 ± 0.00	1.01 ± 0.07
**Pain/discomfort**				
Hospital admission	2.58 ± 0.55	2.05 ± 0.94	2.43 ± 0.5	2.26 ± 0.47
After one month	1.87 ± 0.34	2 ± 0.45	1.57 ± 0.5	1.41 ± 0.49
After one year	1 ± 0.00	1.55 ± 0.60	1 ± 0.00	1 ± 0.00
**Despair**				
Hospital admission	2.89 ± 0.69	2.10 ± 0.71	4 ± 0.77	4.23 ± 0.63
After one month	3.95 ± 0.92	4.70 ± 0.57	2.14 ± 0.57	2.10 ± 0.44
After one year	1.58 ± 0.5	2.75 ± 0.44	1 ± 0.00	1 ± 0.00
**General health state**				
Hospital admission	48.95 ± 9.25	78 ± 12.8	31.9 ± 7.49	29.20 ± 6.64
After one month	30.13 ± 10.55	21.5 ± 8.75	57.14 ± 9.56	59.35 ± 7.02
After one year	83.16 ± 7.74	59 ± 9.67	91.9 ± 7.49	94.46 ± 6.63

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
