# Peer review of "Quality of Life in Patients with Surgically Removed Skin Tumors"

_medicina, 2020, doi:10.3390/medicina56020066_

Round 1

Reviewer 1 Report

Important Topic. Well designed manuscript. Few comments:

Line 63: More details on quality of life. How is quality of life defined? Why is it so important?

Line 103 up to 106: Does it makes sense to measure the diameter of the tumors? Why didn't you measure the time of diagnosis or first apperence until     surgical excision?

Table 2: The unity of Tumor Diameter is missing.  How did you messure the sun exposure?

Line 131: You wrote "CSC"! Do you mean SCC?

Table 3: Unitiy is missing. Also the legend of the table is missing

Line 186: Space is missing

Author Response

Line 63 - We consider that the QoL is in strong connexion with the patients' disease, so it is important to determine if the patients have/have not a fulfilling life. Afterall, the wellbeing of the patients is the most important thing. 

I modify in the manuscript: 

Line 61-65: Both NMSC and MM involve several implications related to high levels of distress or behavioral alteration in patients diagnosed with skin cancer [6], from the impact of the scars to the diagnosis of malignancy. All these can modify the patients’ ability to have a fulfilling life. Moreover, up to 25% of newly diagnosed individuals with neoplasia present symptoms of depression [7].

Line 103 : All the diameters are taken from the histopathological results. The pathologist measure exactly all the tumors. 

Table 2 - I modified in the text.

The sun exposure was evaluated from the patients history - max 1 h in the sun per day - less frequent and more that 3 h - frequent

Line 131 - I corrected it. It was SCC. 

In Table 3 I presented the scores of the QoL between the groups.  What can i improve more? 

Line 186 - corrected it

Reviewer 2 Report

This is an interesting manuscript. However, minor revisions are needed:

-  Please, in the Table  briefly specify which type of BCC you have included in your study (superficial, nodular, nodulo-cystic, sclerodermiform, etc..)

-  Please, in the Table briefly specify which type of AK you have included in your study (KIN I, KIN II or KIN III)

-  Please, in the Table briefly specify which type of melanoma you have included in your study (LM, SSM, ALM, nodular, etc...)

-Please, in the discussion, speculate also about the new surgical treatments and diagnostic items, such as Mohs surgery using Digital ex-vivo confocal, that allows to know the correct margins of the tumor, with a clarity similar to conventional histology and superior to frozen section allowing an immediate evaluation of tumoral margins, reducing unnecessary surgical overtreatment and reducing the aestetic impact on the patients. In this regard add in the reference the article "Digital ex-vivo confocal imaging for fast  Mohs surgery in nonmelanoma skin cancers: An emerging technique in dermatologic surgery. Dermatol Ther. 2019 Nov;32(6):e13127. doi: 10.1111/dth.13127. Epub 2019  Nov 12. PubMed PMID: 31628777."

-Besides, in thie discussion speculate also about non-surgical treatments of NMSCs that can have a reduced impact in the quality of life of patients as reported in the article "Sequential treatment of daylight photodynamic therapy and imiquimod 5% cream for the treatment of superficial basal cell carcinoma on sun exposed areas. Dermatol Ther. 2019 Mar;32(2):e12788. doi: 10.1111/dth.12788". Add this article in your references

-Besides please speculate about the impact of melanoma in head and neck according to facia aesthetic units.

Author Response

-We did not specify the the of BCC and melanoma because the final results for 247 were not anymore significantly statistic.

-The anatomopathologist from our hospital  do not give us the types for AK. 

-Due to the fact that in our clinic the Mohs surgery is not performed I could not speculate about it. 

-All our patients were send by the dermatologists for surgical excision of the lesion. They decided that the nonsurgical treatment was not sufficient. 

-The impact is higher if the melanoma is on the face, due to the remaining scars from the surgical excision, but also from the lymph node dissection.